# A Retrospective Analysis of 10 Years of Liver Surveillance Undertaken in Uveal Melanoma Patients Treated at the Supraregional “Liverpool Ocular Oncology Centre”, UK

**DOI:** 10.3390/cancers14092187

**Published:** 2022-04-27

**Authors:** Alda Cunha Rola, Helen Kalirai, Azzam F. G. Taktak, Antonio Eleuteri, Yamini Krishna, Rumana Hussain, Heinrich Heimann, Sarah E. Coupland

**Affiliations:** 1Liverpool Ocular Oncology Research Group, Department of Molecular and Clinical Cancer Medicine, Institute of System Molecular and Integrative Biology, University of Liverpool, 6 West Derby Street, William Henry Duncan Building, Liverpool L7 8TX, UK; a.rola@liverpool.ac.uk (A.C.R.); h.kalirai@liverpool.ac.uk (H.K.); afgt@liverpool.ac.uk (A.F.G.T.); antonio.eleuteri@liverpool.ac.uk (A.E.); yamini.krishna@liverpoolft.nhs.uk (Y.K.); rumana.hussain@liverpoolft.nhs.uk (R.H.); heinrich.heimann1@nhs.net (H.H.); 2Liverpool Clinical Laboratories, Department of Cellular Pathology, Liverpool University Hospitals Foundation Trust, Liverpool L7 8XP, UK; 3Department of Medical Physics and Clinical Engineering, Royal Liverpool University Hospital, Liverpool L7 8XP, UK; 4Liverpool Ocular Oncology Centre, Liverpool University Hospitals NHS Foundation Trust, Liverpool L7 8XP, UK

**Keywords:** liver surveillance, frequency, modality, detection of metastases, LUMPO3 model, new output of LUMPO3

## Abstract

**Simple Summary:**

Around 45% of patients with uveal melanoma (UM) develop liver metastases on average 3 years after diagnosis of the primary tumour. After clinical detection of metastases, median patient survival is approximately one year. Early identification of metastases through liver surveillance is important so that targeted treatment can benefit affected patients, aiming to prolong their survival. The aim of our retrospective study was to investigate and correlate the characteristics of UM patients diagnosed and treated at a UK supraregional referral center, the Liverpool Ocular Oncology Centre (LOOC), and who were included in the Centre’s liver screening programs for screening for liver metastases. “Real-world” data on the frequency of liver screening in patients after diagnosis and treatment of primary UM are lacking. Through the liver screening program, we found that metastases were detected in 37% of the 615 UM patients studied. A new output based on the prognostic indices of the Liverpool Uveal Melanoma Prognosticator Online version 3 (LUMPO3) model was fitted to the dataset of these patients and accurately estimated the time of onset of metastases.

**Abstract:**

Purpose: To determine liver screening frequency and modality in UM patients following primary treatment, and the characteristics of detected metastases. Methods: A 10-year retrospective study of 615 UM patients undergoing liver surveillance in Liverpool. Information was collected from liver scan reports of these patients. Results: Of 615 UM patients analyzed, there were 337 men (55%) and 278 women (45%). Median age at primary treatment was 61 years (range, 22–94). At study end, median follow-up was 5.1 years, with 375 patients (61%) alive and 240 deceased (39%). Of the deceased patients, 187 (78%) died due to metastatic UM; 24 (10%) deaths were due to other causes; and 29 (12%) patients died of unknown conditions. In total, 3854 liver scans were performed in the 615 UM patients, with a median of 6.2 scans per patient (range, 1–40). Liver MRI was most frequently performed (62.8%). In total, 229 (37%) UM patients developed metastases during the study period: 150 were detected via liver surveillance and 79 were observed post-mortem. Conclusions: Metastatic UM onset is related to the size and genetic profiles of the primary UM, and can be predicted using the model LUMPO3. Regular liver surveillance allowed for timely detection of metastases, and through metastasectomy can lead to prolongation of life in some patients.

## 1. Introduction

Liver metastases are the leading cause of death in patients with UM [1]. Although the success of local control of primary UM with radiotherapy and/or surgery is high [2,3,4,5,6,7], the mortality rate of UM patients is between 40 and 45% due to metastatic disease [7,8,9,10,11]. Only ~2% of UM patients have evidence of metastatic disease at the time of initial diagnosis [12,13]. Although the time of onset of UM metastases varies and even can be up to four decades after diagnosis [14], metastases are usually identified around three years after primary tumor diagnosis [15]. Once liver metastases are clinically detectable, the median patient survival time is less than one year but depends on several factors, including the location and extent of UM metastases as well as any treatments they may have undergone [8,10,16,17,18,19].

Metastatic risk can be determined from certain features of the primary UM. Features associated with a higher risk of developing liver metastases are: large tumor size, ciliary body involvement, extraocular extension and histological characteristics (e.g., the presence of epithelioid cells, connective tissue loops, and a high mitotic count [20,21,22]), chromosomal alterations (e.g., monosomy 3 and chromosome 8q gain) as well as particular gene mutations, most importantly *BAP1* loss [23,24,25,26]. At the Liverpool Ocular Oncology Center (LOOC), individualized patient metastatic death risk is assessed using the Liverpool Uveal Melanoma Prognosticator Online tool (LUMPO3) (www.lumpo.net (accessed on 20 April 2022)); it incorporates clinical, histological, and genetic parameters of each patient’s tumor. This tool has been validated in Liverpool [21], and externally at various centers around the world [27,28,29]. In addition, using the prognostic indices provided by LUMPO3 we recently reported its utility to predict the onset of metastases in UM patients reducing the number of liver screening examinations undertaken, and thus the costs to the National Health Service (NHS) England [30].

Considering the variable time periods during which metastases can occur and the high rates of metastases occurring in the liver in UM patients, it is essential to obtain real-world data regarding liver surveillance to audit patient care and improve their management. Currently, there is no consensus on the imaging modality and frequency for screening UM metastases, and each institution has its own surveillance protocols. Generally, oncologists recommend more intensive surveillance for UM patients with high-risk tumors [31]. In Europe, liver surveillance procedures include ultrasound (US) of the liver performed every 6 months for 10 years, and Magnetic Resonance Imaging (MRI) when a lesion(s) is suspicious of metastatic UM. In the UK, the UM guidelines [32] recommend that high-risk UM patients should have biannual lifelong liver surveillance, which includes liver imaging with MRI and/or US. In the United States of America (USA), the National Comprehensive Cancer Network (NCCN) recommend annual surveillance imaging for low-risk patients; every 6–12 months over 10 years for medium risk, while for high-risk patients, the recommendation is every 3–6 months for 5 years, and then every 6–12 months up to 10 years [33].

Because of the variability in the modalities implemented and the frequency of surveillance undertaken for metastatic disease in UM, as well as the lack of consensus on these measures, the aim of this study was to investigate and correlate the characteristics of UM patients treated at a supraregional referral center in the UK—i.e., LOOC. It included detailed analysis of the liver screening programs at that center, from the time of diagnosis of the primary UM to the time of metastasis detection, in addition to the subsequent patient follow-up.

## 2. Results

### 2.1. Baseline Characteristics of the Patients

Radiological liver screening reports were reviewed and data from 615 UM patients treated at the LOOC were collected. The 615 patients [278 (45%) females, and 337 (55%) males] had a median age of 61 years (range, 22–94). The median primary tumor largest basal diameter (LBD) and ultrasound height (UH) were 14.6 mm (range 2.4–26 mm) and 6.3 mm (range 0.7–20.2 mm), respectively. Patients underwent different types of surgical and non-surgical treatments including enucleations 314 (51%); plaque brachytherapy 131 (21%); proton beam irradiation 118 (19%); local resection 35 (5.7%); endoresection 15 (3%); and endoresection + plaque brachytherapy 2 (0.3%).

Of the 615 cases, 57 (9%) UM had extraocular extension (EOE), and 183 (30%) UM had ciliary body involvement (CBI). Histology was not available for all cases, e.g., in some intraocular biopsies where the cells were very sparse, or where treatment was commenced based on clinical features only. Where histomorphological analysis was possible, 334/578 (58%) tumors showed epithelioid cells, 238/363 (66%) had closed loops, and 369/369 (100%) UM showed between 1 and >7 mitotic counts per 40 high-powered fields.

Genetic information was also not available for all UM, either due to no consent from the patient for genetic testing, or insufficient cellular material for DNA extraction. The following genetic alterations associated with outcome were detected in the cohort: chromosome 1p loss [135/432 (31%)], chromosome 3 loss [330/550 (60%)], chromosome 6p gain [137/432 (32%)], and chromosome 8q gain [257/433 (59%)].

The median follow-up of the 615 patients was 5.1 years (range 0.2–32) (Table 1). Of these, 375 (61%) were alive at the end of this study (01/05/2020). Of the 240 (39%) patients who were deceased, the causes of death were UM metastases 187 (78%); other causes 24 (10%); and unknown, 29 (12%) (Figure 1).

### 2.2. Prognostic Factors Related to the Risk of Metastatic Disease

We analyzed known prognostic factors associated with the risk of developing metastases and undertook Kaplan–Meier survival analyses. As expected, patients with poor outcome in this cohort were associated with known clinical, histological, and genetic features related to UM metastases: increasing tumor size (*p* < 0.001); presence of epithelioid cells (*p* < 0.001); presence of closed loops (*p* < 0.001); high mitotic count (>7) (*p* < 0.001); CBI (*p* < 0.001); EOE (*p* < 0.001); and the genetic alterations-chromosome 3 loss (*p* < 0.001) and chromosome 8q gain (*p* < 0.001) (Appendix A).

### 2.3. Liver Screening Analysis

A total of 3854 liver scans were performed on the 615 patients, with a median of 6.2 scans per patient (range, 1–40). Liver MRI was the most performed modality, 2419 scans (63%). Overall survival (OS) for patients undergoing MRI alone was 23 years (95% CI, 20.2–25.7), followed by 14 years (95% CI, 12.4–15.7) for patients undergoing US only, and 10.2 years (95% CI, 8.6–11.8) for patients who underwent both MRI and US (Figure 2) (Table 2).

In 615 UM patients, 286 (46%) had their first scan at the time of primary tumor treatment, but only 2.4% had metastases detected in that first scan. In total, 229/615 (37%) UM patients developed metastases during the study. Of these, 150 patients (24%) had their metastases diagnosed during the liver surveillance program and a further 79 (13%) were detected on autopsy (Figure 1). In sum, 87% (131/150) of metastases were revealed within 5 years, and 97% (146/150) were revealed up to 10 years after primary tumor treatment.

In the 150 patients with metastases detected by liver screening, a total of 1600 scans were performed; whereas in the 386 patients with no metastases detected during this study, a total of 2114 scans were performed (Appendix A).

### 2.4. Characteristics of Metastatic UM, Survival Analyses, and LUMPO3 Predictions

The median diameter of the largest UM metastases (LDML) was 36 mm (range, 4–196). Metastases categorized according to the AJCC TMN showed; M1a, *n* = 87 (58%); M1b, *n* = 50 (33%); and M1c, *n* = 13 (9%). The median time from primary treatment to the detection of UM metastases was 2.6 years (Range, 0.1–17.8 years). Eighty-one (54%) UM patients undergoing liver screening developed metastases within 2 years of primary tumor treatment; 27 (18%) developed metastases between 2–3 years; and 42 (28%) patients developed metastases 3 years after primary tumor treatment (Figure 3A). The median time from the detection of metastases to death was 1 year (95% CI, 0.7 to 1.3 years). Of the 115 patients who died, 98 (85%) passed ≤2 years after the development of metastatic disease; 11 (9%) died 2–3 years later; and 6 (5%) patients died 3 years following the first detected metastases (Figure 3B). The cumulative incidence for the onset of metastases are shown in Figure 3C and Table 3.

The survival curve showed a median OS of the 150 patients with metastases detected by liver surveillance, of 3.5 years (95% CI, 2.8 to 4.1) (Appendix A). Thirty-five (23%) patients were alive at the end of this study [median OS of 7 years (95% CI, 5.7–8.8)]. (Appendix A). In these 35 patients, the median time to detection of metastases was 4.6 years (95% CI, 3.2–5.8), and the median survival after detection of metastases was 2.5 years (95% CI, 1.8–3.0) (Appendix A).

Kaplan–Meier survival curves were examined for patients categorized according to the three groups defined in Methods—Time to Detection and are shown in Figure 4. Patients from Group 1 [*n* = 108 (17%)] were associated to the poorest outcome—OS time was 2.0 years (95% CI, 1.7–2.2). Overall survival for Group 2 [*n* = 121 (20%)] and Group 3 [*n*= 386 (63%)] were 6.8 years (95% CI, 5.7–7.8) and 26.4 years (95% CI, 24.8–28.0), respectively.

### 2.5. Survival for UM Patients Who Underwent Metastasectomy

In total, 17 (11%) of the 150 patients with metastatic disease underwent metastasectomy. They had a median time to detection of metastases of 3.1 years (95% CI, 1.9–4.3) and a median time to death after UM metastases detection of 1.4 years (95% CI, 0.05–2.8) (Appendix A). The remaining 133/150 (89%) patients where no surgical records were found, had a median time to detection of metastases of 2 years (95% CI, 1.6–2.4), and a median time to death after UM metastases detection of 0.9 years (95% CI, 0.7–1.1) (Appendix A). The median OS for patients undergoing liver resection was 5.9 years (95% CI, 4.6–7.2), and for patients receiving palliative treatment was 3.2 years (95% CI, 2.8–3.6) (Appendix A).

## 3. Discussion

In this extensive retrospective study, we undertook a detailed analysis of radiological reports of 615 UM patients who underwent liver surveillance between 2008–2018 in Liverpool, and correlated the liver screening characteristics with features of the primary tumor as well as with outcome. Our analysis showed that: a) there was a 37% incidence of UM metastases in this cohort; b) the onset of metastases is related to various clinical, histological and genetic parameters of the primary tumor; c) MRI was the most frequently used surveillance technology; d) the median period from primary treatment to UM metastases detection was 2.6 years (95% CI, 1.7–3.9 years); e) median period from UM metastases detection to death was 1 year (95% CI, 0.7–1.3 years); f) the output of the new model based on the prognostic indices of the LUMPO3 model can accurately predict the time of onset of metastases in UM patients; and g) liver surveillance enabled earlier detection of metastases and their surgical removal in some UM patients, leading to slightly prolonged survival.

The incidence of UM metastases in our study of 37% is similar to previously reported data. For example, Diener-West [34] when screening UM patients for metastases, and Rantala [35] when evaluating metastatic UM patients managed with best supportive care, they reported 32% cases of UM metastases in their cohorts of 2320 and 338 patients, respectively. In contrast, other studies report variable data from differing-sized cohorts, with metastases arising in 13–72% of patients [36,37,38,39]. The type of studies performed, and the methods used for detecting UM metastases in the liver may explain this variability between studies.

Our liver surveillance program demonstrated similar results to other national [15] and international studies [39]. Following the UK Guideline recommendations for liver surveillance programs in UM patients [32], the modality of choice in Liverpool is MRI and/or US, with the former technology being used more often in patients with a high-risk of metastasis. In general, MRI is considered to be more specific and sensitive than CT to detect liver UM metastases, whilst US results can be affected by patient body mass indices [40]. However, in our patient cohort, MRI was not always possible on every occasion and, therefore, either US and/or CT was performed. Finnish data employing US detected 48% metastatic UM with 37% visualized by CT and 15% by MRI [39]. In our cohort, 2.4% had metastases detected at first scan. This is similar to Rantala’s study that reported the presence of 4.7% of UM metastases at primary tumor treatment.

From our clinical imaging study, where 54% (81/150) of UM patients developed metastases within 2 years, the overall median time to metastases detection after primary treatment was 2.6 years (range 0.1–17.8 years), and survival was 1 year (95% CI, 0.7–1.3 years) when calculated from the time of detection of liver metastases. We compared the results with previous studies reporting the outcome of metastatic disease of UM, which are summarized in Appendix A. Rietschel et al. [11] evaluated survival parameters in 119 patients with metastatic UM: median time to metastases detection was 4.4 years (range, 0.2–9.9), and median follow-up time was 17 months with 26% of patients alive at 4 years. The estimated median OS was 12.5 months. Similar survival data for metastatic UM were reported by Rivoire et al. [41]: median time to detection of metastases of 29 months, and 87% of patients died within 2 years metastasis detection, compared to 85% in our analysis. They reported a median OS time for survivors of 29 months. Lane et al. [42] described a median OS of only 3.9 months. In contrast to our study, their reported median time from UM primary treatment to detection of metastases was 3.5 years. In total, only 12% of UM patients survived for more than 1 year in their analysis. Lorigan et al. [43] described that in 61% of UM patients metastases were detected 4.3 years after the UM primary treatment. A poor survival time after detection of metastases was observed: 96% of patients died 10 months after diagnosis, compared to 85% who died within two years of diagnosis in our study. Only 3% of patients were alive after detection of metastases, with progressive disease, compared to 23% who survived up to 2.5 years in this analysis.

Seventeen UM patients in our study underwent metastasectomy and had an improved OS compared with patients receiving only palliative care. This agrees with other published series that demonstrated a survival benefit for patients after hepatectomy for UM metastatic disease, when compared to inoperable patients [15,38,44,45,46]. Gomez et al. [38] reported a study of 218 high metastatic risk UM patients enrolled in a liver surveillance program, of which 155 (71.1%) patients had liver metastases detected by clinical imaging: 17 (11.6%) of the patients had metastasectomy. The median OS for patients treated with surgery and ablation was 27 months (range, 14–90), and median OS was 8 months (range, 1–30) in the palliative group. Similarly, Marshall et al. [15] in a study describing metastases detected in 90/188 (48%) high metastatic risk UM reported that 12 patients (13.3%) underwent liver resection with a median OS of 24 months (95% CI, 20.2–27.8 months) compared to 10 months (95% CI, 8.1–11.9 months) in patients with inoperable metastases.

A study published by the Institut Curie [45] described liver screening in 100 high metastatic risk UM patients. Metastases were detected in 60 UM patients, and 50/60 (83%) had only liver metastases. Median OS for all patients in this study after metastases detection was 14 months; however, this increased to 40 months for patients who underwent surgical resection. Another series from Paris [46] reported results of 97 UM patients who underwent hepatic resection, of whom 14% had two successive treatments in the liver: first, liver resection and, second, radiofrequency ablation (RFA). OS after first metastases detection was 70% in 5 years (range 0.49–1.0) and 35% in 10 years (range 0.13–0.92). In comparison, the median OS after RFA was 68% at 2 years (range, 0.47–0.99) and 45% at 4 years (range, 0.23–0.90). These data suggest a survival benefit in some UM patients with liver metastases who underwent RFA in experienced centers.

## 4. Methods

We performed a 10-year retrospective review of liver scan reports of 615 UM patients diagnosed and treated at the LOOC prior to 2018.

### 4.1. Patients

A total of 2254 UM patients were identified from the Ocular Oncology Biobank (OOB), who had given consent to have their health records reviewed for research purposes. The study was approved by both the Health Research Authority (HRA) (NRES REC REF: 18/NW/0748) and the Confidentiality Advisory Group (18/CAG/0181). Of the 2254 UM patients treated at the LOOC, 1448 (64%) cases were excluded since no radiological reports were found for these patients within the records at the Liverpool University Hospitals NHS Foundation Trust (LUHFT): i.e., these patients had their liver surveillance examinations elsewhere in the UK, and hence these data were not accessible/available for review. For 806 (36%) UM patients, radiological reports were found in the LUHFT records; however, in 191 (23%) cases, the radiological examinations were not relevant for the purposes of this study. Therefore, liver scan reports to 615 UM patients, who underwent liver surveillance between 2008–2018, were found and reviewed for this study.

### 4.2. Data Collection

Where available, the following data were collected from the medical records: (1) date of first liver scan; (2) number of liver scans performed; (3) number, size and location of the metastases detected; (4) time from primary treatment to detection of first metastases; and (5) time from detection of metastases to death. After collecting information from the liver scan reports, data was returned to the OOB for pseudo-anonymization of the analysis dataset, including any additional available clinical information: i.e., patient demographics (age/gender); anatomical data of the primary UM—e.g., LBD, UH, CBI, EOE; histological data including presence or absence of epithelioid cells (Epi), presence/absence of connective tissue loops (‘loops’), and mitotic count per 40 high power fields (Mitoc)-; as well as genetic data from tumor cells, including information on the status of chromosomes 1p, 3, 6p, 6q, 8p and 8q (where available). Follow-up and outcome data were also added, where the date of the last follow-up was defined to 01/05/2020. Data from a small number of UM patients analyzed in this study who underwent metastasectomy have been previously published (14, 34); they have been included in this analysis, since we have obtained longer follow-up information.

### 4.3. Categorization of the Primary Tumors

Primary tumors were categorized according to the American Joint Committee on Cancer (AJCC) Tumor Nodal Metastases (TNM) staging (8th edition) (35).

### 4.4. Description of Imaging Modality, Frequency, and Regularity

The imaging modality used and referred to in the surveillance reports were varied and included MRI, US and computer tomography (CT).

### 4.5. Description of the Metastases Found

To classify the size of the metastases, the ‘M’ sub-staging according to the AJCC/TMN staging system for metastatic UM was used (35). Patients were classified into 3 groups according to the diameter of their largest detected liver metastases (M1a < 30 mm, M1b 31–80 mm, and M1c > 80 mm).

### 4.6. Time to Detection

Patients were divided into three groups according to the median time of detection of metastases: Group 1—patients with UM who developed metastases within 2 years of the primary treatment of UM. Group 2—UM patients who developed metastases 2 years after treatment of the primary UM; Group 3—UM patients in whom metastases were not reported during the period of liver screening.

### 4.7. Statistical Analysis

Pseudo-anonymized data were collected, filed, and processed in Excel format (Microsoft, Inc., UoL, London, UK). Categorical variables were summarized with counts and median with range and expressed as frequencies and percentages. A new model in which two predictors (outputs of LUMPO3) were considered: (1) prognostic index of death due to other causes; and (2) prognostic index of death due to metastases; was used to estimate the time of onset of metastases [30]. Statistical analysis was performed using SPSS software V27 (IBM) and checked by coauthors AFGT and AE.

## 5. Conclusions

Our study presents an extensive retrospective analysis of liver surveillance, evaluating its frequency and time interval, in UM patients treated in one of England’s supraregional ocular oncology centers. The new model based on prognostic indices output by LUMPO3 can accurately predict the time of onset of metastases in UM patients. With the LUMPO3 predictions provided, different screening strategies in these patients can be implemented. It has disclosed several noteworthy points despite missing data, particularly that liver surveillance can enable early detection of metastases and potentially, in some patients, lead to prolonged survival benefit through metastasectomy. Therefore, it is necessary to identify UM patients with a high-risk of metastases and excise UM metastases in a timely manner, possibly in combination with other liver-directed treatments. Ongoing basic research may enable the development of adjuvant clinical trials for high metastatic risk UM patients to target UM deposits when there is a low metastatic tumor volume. This study also reveals a need for comparative studies across health systems and causal analysis, to establish consensus for liver screening strategies in UM.

## Figures and Tables

**Figure 1 cancers-14-02187-f001:**
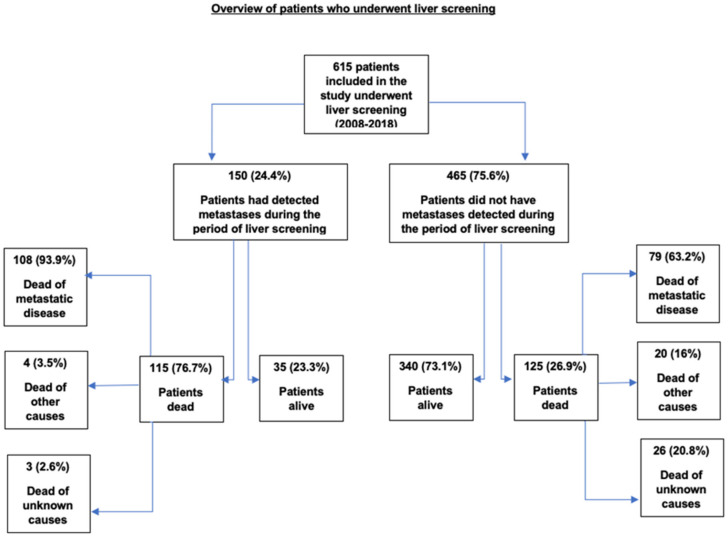
Overview of 615 patients who underwent liver screening according to the detection of metastases, outcome and cause of death.

**Figure 2 cancers-14-02187-f002:**
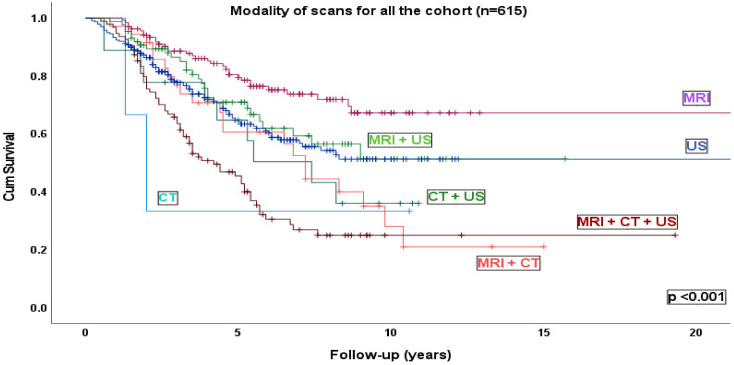
Kaplan–Meier survival curve and table where all the patients were stratified according to modality of scan performed. MRI: 137 patients underwent 983 scans; CT: 3 patients underwent 7 scans; US: 239 patients underwent 515 scans; MRI + CT: 36 patients underwent 400 scans; MRI + US: 87 patients underwent 644 scans; CT + US: 18 patients underwent 84 scans. MRI + CT + US: 95 patients underwent 1223 scans. Number of events indicates the number of deaths.

**Figure 3 cancers-14-02187-f003:**
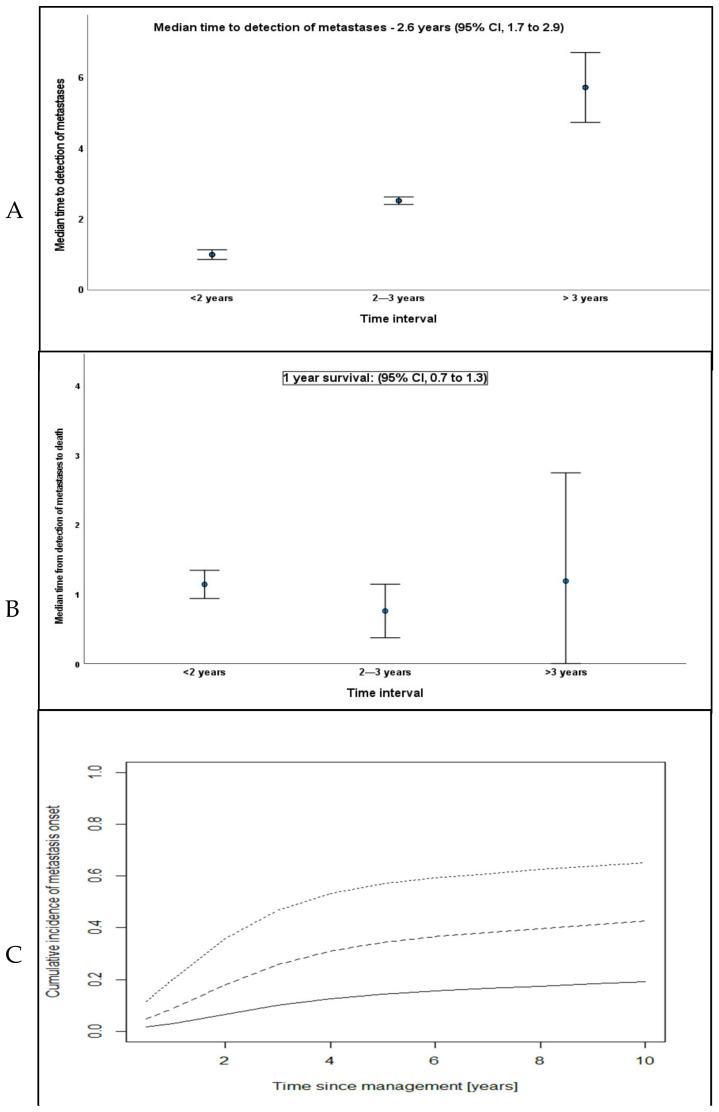
Bar charts with estimates for all 150 patients with metastatic UM from primary treatment to detection of metastatic disease, time from detection of metastatic disease to death, and LUMPO3-based predictions for the cumulative incidence of metastases onset. (**A**) Time from tumor management to detection of metastases for all patients. (**B**) Time from detection of metastases to death. (**C**) Expected cumulative incidences for three risk groups (by LUMPO3 prognostic index of metastatic mortality at 5 years): black line, < 0.69; dashed line, ≥ 0.69 and < 2.07; dotted line: ≥ 2.07.

**Figure 4 cancers-14-02187-f004:**
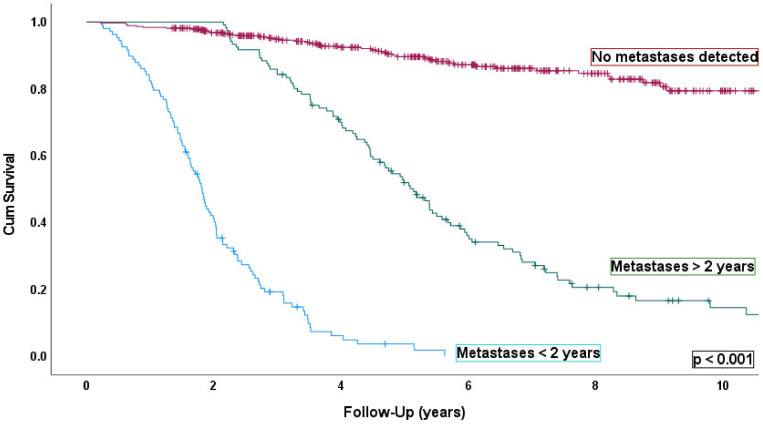
Kaplan–Meier survival curve and table where all the patients were stratified according to when and whether developed metastases; Group1—patients diagnosed with metastases < 2 years (*n* = 108) (*p* < 0.001); Group 2—patients diagnosed with metastases after 2 years (*n* = 121) (*p* < 0.001); and Group 3—patients who had not been diagnosed with metastases at study closure (*n* = 386) (*p* < 0.001). Number of events indicates the number of deaths.

**Table 1 cancers-14-02187-t001:** Patient’s Characteristics, overall data (2008–2018).

Patient’s Features	Total	(%)
**Median age at primary treatment** **(Range)**	61 years (22–94 years)	
**Gender**		
Male	337	55%
Female	278	45%
**Tumors Median Largest Basal Diameter (range)**	14.6 mm (2.4–26 mm)	
**Tumors Median Ultrasound Height (range)**	6.3 mm (0.7–20.2 mm)	
**Ciliary body involvement**		
No	432	70%
Yes	183	30%
**Extraocular melanoma**		
No	558	91%
Yes	57	9%
**Epithelioid cells present**		
No	244	40%
Yes	334	54%
N/A ^1^	37	6%
**Closed PAS+ Loops present**		
No	125	20%
Yes	238	39%
N/A ^2^	252	41%
**Mitotic count**		
0–1	31	5%
2–3	121	20%
4–7	141	23%
>7	76	12%
N/A ^2^	246	40%
**Chromosome 1p**		
Normal	257	42%
Loss	135	22%
Other ^1^	40	6%
N/A ^3^	183	30%
**Chromosome 3**		
Normal	158	26%
Loss	330	54%
Other ^2^	62	10%
N/A ^4^	65	10%
**Chromosome 6p**		
Normal	216	35%
Gain	137	22%
Other ^3^	79	13%
N/A ^3^	183	30%
**Chromosome 6q**		
Normal	293	48%
Loss/Gain	98	16%
Other ^4^	41	6%
N/A ^3^	183	30%
**Chromosome 8p**		
Normal	271	44%
Loss/Gain	129	21%
Other ^4^	32	5%
N/A ^3^	183	30%
**Chromosome 8q**		
Normal	139	23%
Gain	257	42%
Other ^3^	37	6%
N/A ^3^	182	29%
**Follow-up time (years)**		
Median	5.1 years	
Range	(0.2–32 years)	
Status		
Alive	375	61%
Dead	240	39%
**Cause of death**		
Metastatic	187	78%
Other	24	10%
Unknown	29	12%

N/A ^1^ = no biopsy taken. N/A ^2^ = no biopsy taken or not reported. N/A ^3^ = no genetic testing undertaken or only MSA analysis of Chr3 performed. N/A ^4^ = no genetic testing undertaken. Other ^1^ = Gain, unclassified. Other ^2^ = Partial loss, unclassified, allelic imbalance. Other ^3^ = Loss, unclassified. Other ^4^ = Unclassified.

**Table 2 cancers-14-02187-t002:** Type of scans undertaken in the 615 UM patients, and the associated results.

Modality of Scans	Number of Patients	(%)	Number of Scans	(%)	Number of Metastases	(%)	Number of Events	(%)	Censored	(%)	Median Survival (Years)	95% Confidence Interval	
												Lower	Upper
**MRI**	137	(22%)	983	(26%)	30	(13%)	33	(24%)	104	(76%)	23.014	20.278	25.750
**CT**	3	(0.5%)	7	(0.2)	1	(0.4%)	2	(67%)	1	(33%)	4.633	0.000	9.419
**US**	239	(39%)	513	(13%)	56	(25%)	84	(35%)	155	(55%)	14.116	12.455	15.777
**MRI + CT**	36	(6%)	400	(10%)	26	(11%)	21	(58%)	15	(42%)	7.656	5.907	9.404
**MRI + US**	87	(14%)	644	(17%)	25	(11%)	28	(32%)	59	(68%)	10.225	8.641	11.810
**CT + US**	18	(3%)	84	(2%)	7	(3%)	10	(56%)	8	(44%)	6.630	4.790	8.469
**MRI + CT + US**	95	(15%)	1223	(32%)	84	(37%)	62	(65%)	33	(35%)	7.375	5.777	8.974
**Overall**	615		3854		229	(37%)	240	(39%)	375	(61%)	17.067	15.576	18.557

**Table 3 cancers-14-02187-t003:** Grouping of the 615 UM patients based on survival ≤2 years, between 2-3 years and >3 years of onset of metastasis.

Groups	Total Number of Patients (%)		Number of Events %		Censored %		Sig.	Median Survival (Years)	95% Confidence Interval	
									Lower	Upper
1	108	17.6%	100	92.6%	8	7.4%	*p* < 0.001	2.011	1.791	2.232
2	121	19.6%	94	77.7%	27	22.3%	*p* < 0.001	6.810	5.778	7.841
3	386	62.8%	46	11.9%	340	88.1%	*p* < 0.001	26.484	24.847	28.071
Overall	615	100%	240	30.3%	375	61.1%		5.065	15.440	18.471

## Data Availability

3rd Party Data. Restrictions apply to the availability of these data. Data was obtained from the Ocular Oncology Biobank (OOB), University of Liverpool and are available from the custodian of the OOB, with the permission of the University of Liverpool pending appropriate ethical approvals and Data Transfer Agreement.

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
