# Peer review of "A Retrospective Analysis of 10 Years of Liver Surveillance Undertaken in Uveal Melanoma Patients Treated at the Supraregional “Liverpool Ocular Oncology Centre”, UK"

_cancers, 2022, doi:10.3390/cancers14092187_

Round 1

Reviewer 1 Report

This study investigated the liver screening frequency and modality in UM patients following primary treatment, and the characteristics of detected metastases at the Liverpool Ocular Oncology Centre (retrospective analysis of 10 years, 615 patients). They showed among others that their new model based on the prognostic indices of the LUMPO3 model can accurately predict the time of onset of metastases, and that the liver surveillance enabled earlier detection of metastases and their surgical removal, leading to slightly prolonged survival for patients that underwent metastasectomy.

Comments:

Line 26: Please define the abbreviation « LUMPO3 » in the Simple summary.

Lines 59-60: Add a bracket after “high mitotic count [22-24]”.

Lines 99-101: The sum of the percentages of this sentence is over 100%. Correct for “enucleations 314 (51%)”, “proton beam irradiation 118 (19%)” and “local resection 35 (5.7%)”.

Line 112: Correct for “chromosome 3 loss [330/550 (59%)]”.

Line 118, Table 1: Percentages for gender are inversed. Put notes below the table in superscript (N/A1, etc; Other1, etc).

Line 129: Correct for “8q gain (p<.001)”.

Lines 130-135: Please indicate values for CT scans (CT, MRI+CT, etc) included in Figure 2.

Line 136: The numbers of the last column of the table included with the Figure 2 are missing (95% confidence interval: Lower/Upper).

Line 141, Figure 2 legend: Add the number of scans for MRI + CT + US.

Line 165, Figure 3: The resolution of this figure could be improved.

Line 187-189: Correct for “(n=108) (p<0.001)”; “(n=121) (p<0.001)”; “(n=386) (p<0.001)”.

Figure S1: Add the p-value for the tumour size panel.

Figure S2 legend: Correct for “Kaplan-Meier survival curve and table where 150 UM patients who developed metastases were stratified according to modality of scan performed.”

Figure S3 legend: Correct for “Kaplan-Meier survival curve and table where 386 UM patients who never demonstrated metastases were stratified according to modality of scan performed.”

Figures S4 and S5: The resolution of these figures could be improved.

Author Response

Thank you very much for your useful comments; the alterations have been highlighted in yellow in the text, figures and figure legends.

Line 26: Please define the abbreviation « LUMPO3 » in the Simple summary. Done.

Lines 59-60: Add a bracket after “high mitotic count [22-24]”. Done.

Lines 99-101: The sum of the percentages of this sentence is over 100%. Correct for “enucleations 314 (51%)”, “proton beam irradiation 118 (19%)” and “local resection 35 (5.7%)”. Done .

Line 112: Correct for “chromosome 3 loss [330/550 (59%)]”. Done.

Line 118, Table 1: Percentages for gender are inversed. Put notes below the table in superscript (N/A1, etc; Other1, etc). Done

Line 129: Correct for “8q gain (p<.001)”. Done.

Lines 130-135: Please indicate values for CT scans (CT, MRI+CT, etc) included in Figure 2. Done.

Line 136: The numbers of the last column of the table included with the Figure 2 are missing (95% confidence interval: Lower/Upper).

Please see Figure S3 (supplementary matter) – Page 2: "Kaplan-Meier survival curve and table where 386 UM patients who never demonstrated metastases were stratified according to exam modality performed".

Please note that in Figure 2, there are no numbers in the last column because for this group of patients who never showed metastases, there are no survival statistics because all cases are censored, and this is already mentioned in the legend.

Line 141, Figure 2 legend: Add the number of scans for MRI + CT + US. Done.

Line 165, Figure 3: The resolution of this figure could be improved.

Done. Figure 3, Page 12. To improve the resolution, the layout of the figure was changed from horizontal to vertical.

Line 187-189: Correct for “(n=108) (p<0.001)”; “(n=121) (p<0.001)”; “(n=386) (p<0.001)”. Done – see Figure 4, page 14 - legend, lines 3,4, and 5.

Figure S1: Add the p-value for the tumour size panel. Done.

Figure S2 legend: Correct for “Kaplan-Meier survival curve and table where 150 UM patients who developed metastases were stratified according to modality of scan performed.” Done - Page 4, lines 1,2, and 3.

Figure S3 legend: Correct for “Kaplan-Meier survival curve and table where 386 UM patients who never demonstrated metastases were stratified according to modality of scan performed.” Done. Page 5, lines 1, and 2.

Figures S4 and S5: The resolution of these figures could be improved. Done – Pages 6, and 8

Reviewer 2 Report

The same as for the Editors.

Author Response

Reviewer 2 wrote "The same as for the Editors." - we could not access any further comments/suggestions and so have not addressed any here.

Reviewer 3 Report

The authors evaluated liver screening frequency and modality in patients affected by primary enucleated uveal melanoma and the characteristics of detected metastases by performing a 10-year retrospective study of 615 UM patients. 

They found that metastases occurred in 229 UM patients during the study period: 150 were detected by liver surveillance and 79 were observed post-mortem. In addition they found that UM metastatic risk may be predicted using the model LUMPO3. The study is interesting. The manuscript is well written and the methods and results well described too.   I have some minor concerns that must be addressed to improve the paper: 1. Genetic landscape of UM should be better described and detailed, as well as the prognostic value of selected immune-escape genes. The following articles may be used: 10.1371/journal.pone.0210276 2. Current therapeutic strategies of UM are not well described and discussed in the present paper.

Author Response

Thank you very much for your suggestions and comments. We have highlighted the alterations in the text, figures and figure legends.

I have some minor concerns that must be addressed to improve the paper:

  1. Genetic landscape of UM should be better described and detailed, as well as the prognostic value of selected immune-escape genes. The following articles may be used: 10.1371/journal.pone.0210276 and 10.3390/app10228081.

Done – included in the Introduction.

  1. Current therapeutic strategies of UM are not well described and discussed in the present paper. The following doi may be used: 10.1038/srep44564.

Done – included in the Introduction.